# Absence of Neuroplastic Changes in the Bilateral H-Reflex Amplitude following Spinal Manipulation with Activator IV

**DOI:** 10.3390/medicina58111521

**Published:** 2022-10-25

**Authors:** Alma Fragoso, Brayan Martínez, María Elena Ceballos-Villegas, Elizabeth Herrera, Juan José Saldaña, Ana Lilia Gutiérrez-Lozano, Elías Manjarrez, Joel Lomelí

**Affiliations:** 1Escuela de Quiropráctica, Universidad Estatal del Valle de Ecatepec, Ecatepec de Morelos 55210, Mexico; 2Escuela Nacional de Medicina y Homeopatía, Instituto Politécnico Nacional, Ciudad de México 07738, Mexico; 3Escuela Nacional de Ciencias Biológicas, Instituto Politécnico Nacional, Ciudad de México 11350, Mexico; 4Instituto de Fisiología, Benemérita Universidad Autónoma de Puebla, Puebla 72570, Mexico; 5Escuela Superior de Medicina, Instituto Politécnico Nacional, Ciudad de México 11340, Mexico

**Keywords:** H-wave, plasticity, neuroplasticity, bilateral H-reflex, M-wave, alpha-motoneuron, monosynaptic reflex, chiropractic, spinal manipulation

## Abstract

*Background and Objectives:* Chiropractic spinal manipulation is an alternative medical procedure for treating various spinal dysfunctions. Great interest exists in investigating its neuroplastic effects on the central nervous system. Previous studies have found contradictory results in relation to the neuroplastic changes in the H-reflex amplitude as a response to manual spinal manipulation. The discrepancies could be partly due to differences in the unilateral nature of these recordings and/or the variable force exerted in manual techniques applied by distinct chiropractors. Concerning the latter point, the variability in the performance of manual interventions may bias the determination of the significance of changes in H-reflex responses derived from spinal manipulation. To investigate such responses, a chiropractic device can be used to provide more precise and reproducible results. The current contribution aimed to examine whether spinal manipulation with an Activator IV instrument generates neuroplastic effects on the bilateral H-reflex amplitude in dancers and non-dancers. *Materials and Methods:* A radiograph verified spinal dysfunction in both groups of participants. Since there were significant differences between groups in the mean Hmax values of the H-reflex amplitude before spinal intervention, an assessment was made of the possible dependence of the effects of spinal manipulation with Activator IV on the basal conditions. *Results:* Ten sessions of spinal manipulation with Activator IV did not cause statistically significant changes in the bilateral H-reflex amplitude (measured as the Hmax/Mmax ratio) in either group. Furthermore, no significant difference was detected in the effects of spinal manipulation between groups, despite their distinct basal H-reflex amplitude. *Conclusions:* Regarding the therapeutic benefits of a chiropractic adjustment, herein carried out with Activator IV, the present findings suggest that the mechanism of action is not on the monosynaptic H-reflex pathway. Further research is needed to understand the mechanisms involved.

## 1. Introduction

The vertebral column (spine) encloses the spinal canal, which in turn contains the spinal cord. The spinous process and the corresponding paraspinal muscles are found at the most dorsal aspect of the vertebral column. These muscles cover a series of facet joints between the inferior and superior articular processes of two adjacent vertebrae. For spinal manipulation, the joints and muscles of the vertebral column are subjected to a controlled, sudden force produced by a chiropractor in a specific direction, in such a manner as not to damage the spinal cord. This technique is often applied manually, causing the amount of force exerted to vary. However, it is also possible to perform spinal manipulations with mechanical devices, such as Activator IV, to improve control over the force exerted in a spinal adjustment [1,2].

The effects of spinal manipulation have been debated for many years [3,4,5,6,7]. According to accumulating evidence, a chiropractic adjustment can produce neuroplastic effects on the central nervous system [8,9], although the physiological mechanisms of these changes are still unclear [9,10]. Regarding H-reflexes following spinal manipulation, there have been contradictory reports. For instance, manual spinal manipulation has been described to decrease the unilateral H-reflex amplitude in individuals with low back pain [11] and in those without pain [12,13,14,15,16]. In contrast, another study found no significant changes in the unilateral H-reflex amplitude in healthy volunteers after receiving spinal manipulation at the S1 joint [11]. Likewise, the application of spinal manipulation to Taekwondo athletes with subclinical pain [17] or to healthy young subjects with spinal dysfunction did not generate significant changes in the area below the unilateral H-reflex recruitment curves (Harea/Mmax) [18]. 

One study examined the effects of spinal manipulation on intersubject variability. Whereas 6 participants showed a reduction in the H-reflex amplitude subsequent to spinal manipulation, 13 others did not exhibit any changes in this parameter [19]. According to the authors, the discrepancy might be due to differences in autonomic tone or technical issues in the spinal manipulation technique. As a more general conclusion, spinal manipulation can produce an increase in the H-reflex amplitude in some individuals and a decrease in others.

Discrepancies between reports could be partly attributed to variations in the application of the manual techniques by distinct chiropractors. To obtain more consistent and reliable data, the force applied on the structures treated was performed with an Activator IV, which had a standardized force graded in four levels, and the effects on bilateral H-reflex responses were evaluated. 

The aim of the current contribution was to determine whether ten sessions of controlled spinal manipulation carried out with Activator IV caused a significant change in the H-reflex amplitude (measured as the bilateral Hmax/Mmax ratios and the bilateral H-reflex recruitment curves) in a dancer and/or non-dancer group. There were significant differences between groups in the initial mean Hmax amplitude. These two groups were chosen because of previous evidence of more limited H-reflexes in dancers versus non-dancers [20]. 

Since spinal manipulation did not modify the H-reflex amplitude in either group, the initial value of this parameter did not affect the results produced by the technique. Moreover, the present findings question the reported existence of a neuroplastic effect of chiropractic spinal manipulation on the segmental monosynaptic H-reflex pathway. Hence, valuable insights are herein provided into one spinal mechanism that is apparently not affected by chiropractic intervention. Such information should be useful for chiropractic researchers in their attempt to clarify the mechanism responsible for the therapeutic effects of spinal manipulation.

## 2. Materials and Methods

### 2.1. Inclusion/Exclusion Criteria

#### Subjects

Fourteen individuals were divided into two groups. The first group consisted of seven regional dancers (mean age: 33 ± 17.5 years) and the second included seven volunteers without experience in dance (mean age: 27.7 ± 15.8 years). The members of the dancer and non-dancer groups were chosen to represent a similar health condition, age range, body mass index, and gender mix. They all had spinal dysfunction (verified by radiography) but without comorbidities, pain, or contraindications to spinal manipulation. All subjects signed informed consent to participate in the study once it was fully explained, in accordance with the Declaration of Helsinki. The protocol was approved by the local ethics committee of the Medical School of the National Polytechnic Institute (SIP protocol 20194947). Full methodological details are provided in the Appendix A to aid in the attainment of reproducible results. Individuals with spinal dysfunction that had comorbidities, pain, or contraindications to spinal manipulation were excluded.

### 2.2. Medical History and X-ray Analysis

A complete medical history was made for each participant as well as an examination of possible alterations in posture, load distribution, and movement. An X-ray was taken of the spinal column of all subjects to verify the initial condition and obtain a concrete idea of the specific sites in need of spinal manipulation. The results of these studies are shown in Table A1 (Appendix B).

### 2.3. Intervention: Spinal Manipulation with Activator IV

Activator IV is a spring-loaded and hand-held mechanical instrument commonly used by chiropractors to deliver a controlled force impulse of 0.3 J to spinal joints (Figure 1A, right panel) to treat neck and back pain. Compared to manual spinal manipulation, the device produces an optimal spine alignment in a shorter time per session.

Before spinal manipulation, each participant was examined in order to program the individual necessities for each session (Table A2, in Appendix B). In all cases, a certified chiropractor followed the guidelines published by Fuhr (2008) [21] for the use of Activator IV. While the participants were in a prone position, spinal manipulation was carried out with Activator IV two times per week for five weeks. A single certified chiropractor performed all spinal manipulations with the same device to avoid the variability implied with multiple operators and/or distinct devices.

### 2.4. Stimulation and the Recording of Parameters to Determine H-Reflexes and M-Waves

Up to 120 rectangular pulses (1 ms) were delivered in 12 min, with a stimulation time interval of 6 s (i.e., 0.166 Hz) to obtain the recruitment curves. Each point on the curve depicts the average of three trials. Intensities were limited to a range of 2.0 to 25 mA for dancers and 0.8 to 32 mA for non-dancers. The stimuli were applied with a Digitimer (DS5) stimulator commanded by a Master-8 stimulator, which generated a synchronous pulse to activate the data acquisition system. The H-reflex and M-wave signals were amplified 500 times with GRASS Astro-med LP511 amplifiers and band-pass-filtered from 10 Hz to 1 kHz. A Digidata 1440 A interface (Axon Instruments, Molecular Devices, Silicon Valley, CA, USA) was utilized with a sampling rate of 50 kHz. Subsequently, the amplitude (peak to peak) of the H-reflex and M-wave was measured as illustrated in Figure 1A with Axoscope software (Molecular Devices, Silicon Valley, CA, USA).

The stimuli to produce the right and left soleus H-reflexes were bilaterally applied to the posterior tibial nerves. The current protocol is similar to one previously reported (Ceballos-Villegas et al., 2017). Recruitment curves were constructed from the data obtained from each individual before (in control conditions) and after undergoing the 10-session intervention, allowing for the visualization of bilateral Hmax and Mmax values. All experiments were carried out from 10 am to 2 pm to avoid circadian variations and thus attain comparable H-reflex amplitudes [22]. 

### 2.5. Statistical Analysis

For each participant, data on the bilateral H-reflex and M-wave were generated before and after spinal manipulation. Based on a comparison of the two data sets, the presence or absence of significant differences was established. For this purpose, the maximum responses for Hmax, Mmax, the Hmax/Mmax ratio, and the area under the H-reflex recruitment curves were employed. These values were previously tested for normal distribution (Kolmogorov–Smirnov) and homogeneity of variance (Levene). The changes in values from pre- to post-spinal manipulation were analyzed with one-way ANOVA. Because some data were not normally distributed (*p* < 0.05), Kruskal–Wallis one-way ANOVA on ranks was also applied. Statistical significance was considered at *p* < 0.05 for all comparisons. Data are herein expressed as the mean ± standard deviation (SD), except in Figure 1, where they are denoted as the mean ± standard error (SE). 

## 3. Results

Firstly, the mean right- and left-Hmax amplitudes existing prior to spinal manipulation (basal conditions) were compared between dancers and non-dancers, finding values of 1.16 ± 0.92 and 3.05 ± 0.81, respectively, for dancers and of 2.85 ± 0.92 and 4.57 ± 1.24, respectively, for non-dancers. The differences between groups are significant for the basal Hmax values (total DF = 27, F = 13.8, *p* < 0.001). These results are consistent with a previous publication [20].

Secondly, to calculate Mmax and Hmax, bilateral recruitment curves were systematically constructed for all participants. They were interpolated with sigmoidal and Gaussian equations of three parameters, respectively. Each recruitment curve was then normalized to Mmax and maximal stimulation intensity. Figure 1 portrays the overall average of recruitment curves for each group in regard to right–left M-waves and H-reflexes before and after spinal manipulation with Activator IV. The superimposed curves display a similar qualitative profile. A quantitative analysis confirmed that there are not any statistically significant differences in the area below the H-reflex recruitment curves between the two measurement times for dancers (F = 0.40, *p* = 0.75) and non-dancers (F = 1.13, *p* = 0.36).

### 3.1. Dancers

Spinal manipulation of dancers with Activator IV caused no significant change in the right and left Hmax (H = 0.56, *p* = 0.90), the right and left Mmax (F = 1.09, *p* = 0.37), or the right and left Hmax/Mmax ratio (F = 0.32, *p* = 0.8). Figure 2A,B clearly illustrates the absence of significant differences between the two measurement times.

### 3.2. Non-Dancers

Likewise, spinal manipulation of non-dancers with Activator IV brought about no significant change in the right and left Hmax (F = 0.11, *p* = 0.95), the right and left Mmax (H = 2.19, *p* = 0.53), or the right and left Hmax/Mmax ratio (H = 2.07, *p* = 0.55). Figure 2C,D portrays the absence of significant differences between the two measurement times.

The results indicate that spinal manipulation with Activator IV did not produce neuroplastic effects on the H-reflex amplitude in either of the two groups (dancers or non-dancers), even though their basal Hmax values were significantly different. Tables with raw data were included in the Appendix A.

## 4. Discussion

The first contribution of the present research was to use a reproducible form of spinal manipulation, which was achieved with an Activator IV instrument. Previous studies have been carried out with manual chiropractic manipulations (without instruments), which certainly generated variability in the force exerted and consequently in the results of the participants. 

Secondly, it was demonstrated that the effects of spinal manipulation are not dependent on the initial mean H-reflex amplitude, a parameter found to be significantly different between groups before the spinal intervention. The latter finding is in agreement with a prior study [20]. 

Finally, no significant change was detected in the bilateral Hmax/Mmax ratio for either group after ten sessions of spinal manipulation with Activator IV. Hence, the therapeutic effects of a chiropractic adjustment, herein carried out with the standardized force of Activator IV, are apparently elicited through a mechanism distinct from the monosynaptic H-reflex pathway. 

The plastic changes caused by therapy may be due to motor tasks involving the activation of descending pathways. Such seems to be the case in studies revealing that the V-wave amplitude can be lengthened by spinal manipulation [17,18]. This idea is consistent with a recent investigation on stroke patients subjected to a chiropractic intervention. They exhibited a greater V-wave amplitude but no significant difference in the H-reflex amplitude [23]. However, the V-wave (cortical drive) depends on the maximum voluntary contraction (MVC), a learning-dependent parameter linked to the motivation of the subject. Thus, it is difficult to attribute observed changes in the V-wave amplitude only to the Ia-afferent motoneuron segmental pathway. 

On the other hand, it is well-known that variation in H-reflex amplitude does not always correspond to a net effect on motoneuron excitability [24], probably because presynaptic inhibition can alter neurotransmitter release on the Ia motoneuron pathway [25,26]. Similarly, the fact that the H-reflex was not different after spinal manipulation does not imply the absence of modifications in the motoneuron excitability. This idea is consistent with the capacity of spinal manipulation to produce a significantly lower H-reflex threshold without any substantial change in the H-reflex amplitude [18]. As a plausible explanation for the latter finding, spinal manipulation could possibly cause greater presynaptic inhibition, which might be balanced by a concomitant increase in the excitability of motoneurons. Such a balance would give rise to the absence of a net effect on the H-reflex amplitude (as the current results indicate), while the motoneurons would be ready to generate plastic changes by the action of descending drives. This interpretation may explain why Niazi et al. (2015) [18] observed a substantial decrease in the H-reflex threshold and an increase in the V-wave amplitude evoked in the soleus [17,18]. Nevertheless, the aforementioned interpretation must be considered cautiously until it undergoes further experimental testing. 

Also of relevance is that the increment in the MVC stemming from spinal manipulation lasted for at least 30 min [18]. Conversely, previous studies found that the attenuation of the amplitude of the H-reflex lasted for only 30 s after the spinal intervention [13], putting in question the clinical relevance of the latter change in relation to long-term motor effects. Therefore, the long-term clinical improvements derived from spinal manipulation are apparently associated with other mechanisms not involving the segmental monosynaptic H-reflex pathway.

### 4.1. Limitations of the Study

The first limitation of the present investigation is the small number of participants in each group. The second limitation is that we did not examine how spinal manipulation with Activator IV may increase presynaptic inhibition concomitantly with a boost in the excitability of motoneurons. Future research is required to explore this possibility.

### 4.2. The Originality of the Study

The results confirm the implicit observations by Niazi et al. (2015) [18] that the H-reflex amplitude does not change after spinal manipulation. The originality of the current contribution lies in extending such observations to bilateral H-reflexes and finding that the absence of a change in the H-reflex amplitude is not dependent on its initial level. 

Indeed, the ability to evaluate participants with distinct basal levels of the H-reflex amplitude was the reason for deciding to work with dancers and non-dancers. H-reflexes were previously found to be smaller in dancers from The Royal Danish Ballet than in well-trained athletes [20], thus suggesting a difference in the H-reflex amplitude between these two groups. Presently, the H-reflex amplitude was also significantly lower in dancers than non-dancers (see Figure 2), in agreement with the report by Nielsen et al. (1993) [20]. No significant differences were detected in the H-reflex amplitude within either group of participants when comparing the measurements before and after spinal manipulation.

In the current contribution, moreover, an evaluation was made of the possible effect of ten sessions of spinal manipulation (versus the single session in the study by Niazi et al., 2015 [18]) on the H-reflex amplitude. A greater number of interventions were herein performed to see whether there was an effect that did not manifest itself with a single spinal intervention. Unlike in previous studies, the present spinal manipulations were carried out with a mechanical device (Activator IV) with the aim of reducing the variability in the results due to manual spinal adjustments. Interestingly, reports show that not all manipulative techniques have the same effect. Differences may be mediated by neurological or biomechanical factors related to each chiropractic technique [27]. Future investigation is necessary on the complexity of bilateral H-reflexes to clarify whether other variables of the H-reflex or M-wave are affected by the chiropractic intervention [28].

## 5. Conclusions

Ten sessions of chiropractic spinal manipulation with Activator IV did not change the H-reflex amplitude in either the dancer or non-dancer group. Hence, the basal level of the H-reflex amplitude, a parameter found to be significantly different between groups, did not affect the outcome. According to the current results, the therapeutic effects of a chiropractic adjustment, herein carried out with Activator IV, are elicited through a mechanism distinct from the monosynaptic H-reflex pathway. Further research is needed to understand the mechanisms by which such therapeutic benefits are obtained. 

## Figures and Tables

**Figure 1 medicina-58-01521-f001:**
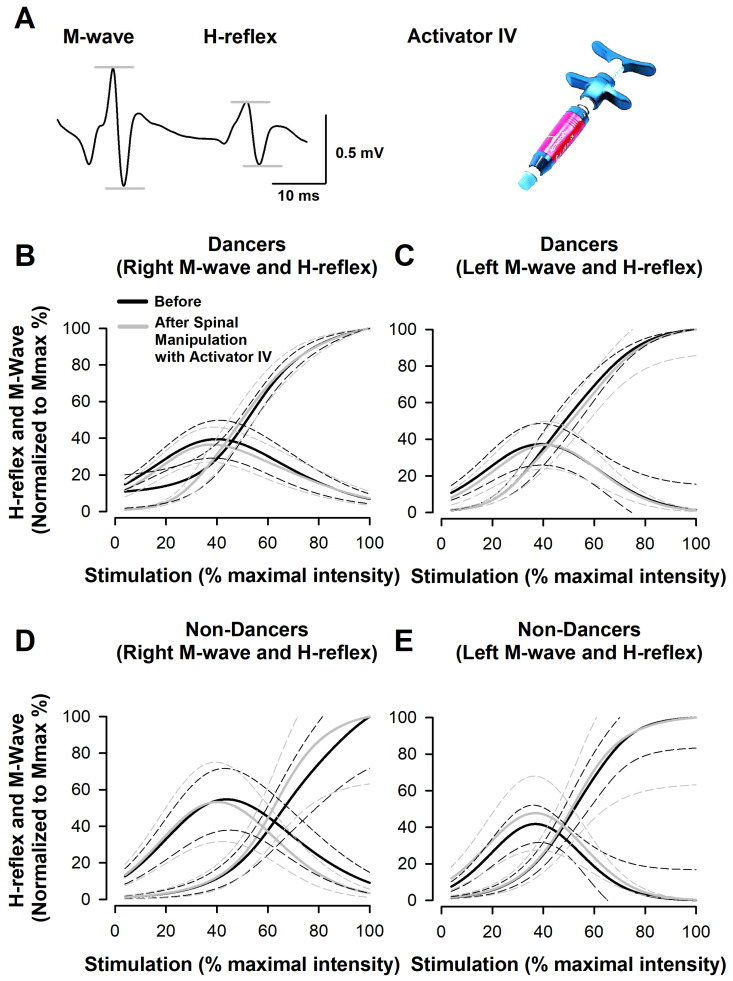
(**A**) Typical recording of an M-wave and H-reflex, and an illustration of the Activator IV instrument. (**B**–**E**) Pooled data. Overall average of fitted recruitment curves for M-waves and H-reflexes recorded in dancers: (**B**) from the right soleus muscle and (**C**) from the left soleus muscle. (**D**,**E**) The same as (**B**,**C**) but for non-dancers. M-wave data were best fitted by a sigmoid nonlinear regression model of three parameters. On the other hand, H-reflex data were best fitted by a Gaussian nonlinear regression model of three parameters. Solid lines portray the overall average and dashed lines depict the standard error.

**Figure 2 medicina-58-01521-f002:**
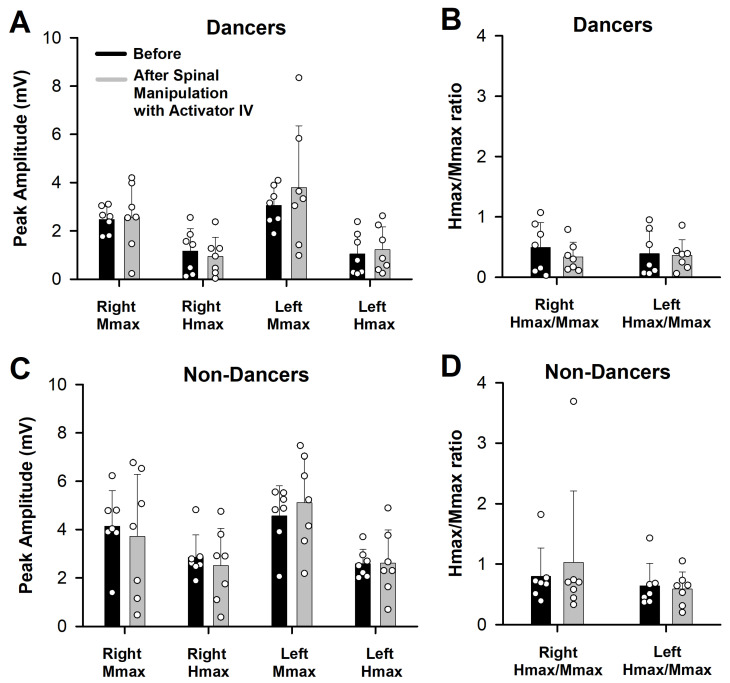
An absence of changes in bilateral Mmax, Hmax, and Hmax/Mmax was found for dancers (**A**,**B**) and non-dancers (**C**,**D**) after spinal manipulation with Activator IV. Bar charts are the mean values. Vertical bars indicate the standar deviation. The white circles illustrate raw data obtained from all participants.

## Data Availability

Not applicable.

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
