# Peer review of "Absence of Neuroplastic Changes in the Bilateral H-Reflex Amplitude following Spinal Manipulation with Activator IV"

_medicina, 2022, doi:10.3390/medicina58111521_

Round 1
Reviewer 1 Report
In this manuscript, the authors investigate neuroplastic effects of chiropractic spinal manipulation on the central nervous system. Previous studies have found contradictory results about them. The authors examine whether spinal manipulation with an Activator Ⅳ instrument generates neuroplastic effects on the bilateral H-reflex amplitude in dancers and non-dancers. The authors’ results suggest that the therapeutic benefits of chiropractic adjustment are not due to the action on the monosynaptic H-reflex pathway.
Comments
The data presented in this manuscript are of potential interest, and the experimental strategies are well designed. One minor point is as follows. Because the values of peak amplitude and Hmax/Mmax (Figure 2) are wide range, the authors should concretely describe about health condition, BMI and gender of the member.
Author Response
The data presented in this manuscript are of potential interest, and the experimental strategies are well designed. One minor point isas follows.
Because the values of peak amplitude and Hmax/Mmax (Figure 2) are wide range, the authors should concretely describe about health condition, BMI and gender of the member.
Thanks for the observation.
All together experimental subjects were neurologically healthy, but all had subluxations that became apparent with the X-ray studies shown in Table A1. In fact, each subject had to have some degree of subluxation to be subjected to the therapy, otherwise they did not enter the study protocol. Chiropractic therapy allows to align the spine which produces clinical improvement in patients.
The attached table shows the characteristics of the study subjects.
|
Subject |
|
|
|
|
|
|
No dancer |
Gender |
Age (years) |
Height (m) |
Weight (Kg) |
BMI |
|
1 |
M |
31 |
1.74 |
74 |
24.4 |
|
2 |
M |
22 |
1.80 |
85 |
26.2 |
|
3 |
F |
21 |
1.60 |
66.7 |
25.8 |
|
4 |
F |
22 |
1.61 |
86.3 |
33.2 |
|
5 |
F |
23 |
1.70 |
61.3 |
21.1 |
|
6 |
F |
60 |
1.55 |
55.0 |
22.9 |
|
7 |
F |
22 |
1.55 |
50.6 |
21.2 |
|
Dancer |
|
|
|
|
|
|
1 |
M |
32 |
1.65 |
66 |
24.2 |
|
2 |
M |
22 |
1.83 |
88 |
26.3 |
|
3 |
F |
21 |
1.60 |
65 |
25.4 |
|
4 |
F |
50 |
1.60 |
83 |
32.4 |
|
5 |
F |
23 |
1.60 |
56.5 |
21.9 |
|
6 |
F |
60 |
1.54 |
56 |
23.6 |
|
7 |
F |
22 |
1.50 |
59.9 |
26.7 |
Reviewer 2 Report
The manuscript entitled” Absence of Neuroplastic Changes in the Bilateral H-Reflex Amplitude Following Spinal Manipulation with Activator IV” by Fragoso et al describes the result of chiropractic spinal manipulation by an Activator IV to measure the bilateral H-reflex amplitude in dancers and non-dancers. Ten sessions of spinal manipulation with Activator IV didn’t change the bilateral H-reflex amplitude. Their finding reveals that therapeutic benefits of a chiropractic adjustment by Activator IV having no involvement of monosynaptic H-reflex pathway. Though, this experiment involving Activator IV provides some interesting observation but it needs to have some more supporting/specific experiments to prove their claim that prevent this paper from being publishable at this stage.
Major concerns:
1. Page 2, line 72, “the force exerted was herein standardized with an Activator IV…..”, How this has been achieved? Include detail experiment procedures with optimization steps to better understand the technique. Is it completely a new experiment (use of Activator IV) or modification of previous techniques?
2. What is the novelty of your finding? Previous finding by Niazi et, al (2015) mentioned that there is no change in H-reflex response but the H-reflex pathway can be infected by spinal manipulation.
3. Page 2, line 51, “Activator IV fixes the force at 0.3 J [1,2]”. Is it the maximum force you can reach or you use this force in your experiment to measure the H-reflex change. Also, the references are not from recent work and not involving the set-up of this work as well?
4. Figure 1B-E, Why the dash lines are always higher from Average value, are the authors showing only the higher limit of standard deviation?
5. Figure 2 shows the peak amplitude and Hmax/Mmax ratio but the no. samples are not the same, it carries between 5-7, why?
6. Overall, the Authors should performed some more relevant experiment using Activator IV for better understanding the H-reflex pathway by spinal manipulation
Minor:
1. Double numbering of references
Author Response
Thank you very much, we fully appreciate your comments and recommendations.
Major concerns:
- Page 2, line 72, “the force exerted was herein standardized with an Activator IV…..”, How this has been achieved? Include detail experiment procedures with optimization steps to better understand the technique. Is it completely a new experiment (use of Activator IV) or modification of previous techniques?
Response: We changed the text for “the force applied on the structures treated was performed
with an activator IV which had a standardized force graded in 4 levels…” Actually, Activator IV was already used and standardized prior to its introduction in clinical practice (Fuhr, 2008). The maneuvers applied and the structures adjusted in our work for each experimental subject are shown in detail in Table A2. In essence, with the activator IV, a (calibrated) blow is applied punctually to the tissue to be treated, for example: in a vertebra (cervical, dorsal or lumbar) the blow is applied in order to align it with the rest of the spine. Our protocol included 10 sessions which is clinically considered complete chiropractic therapy. This is the first time that an analysis of the electrophysiological effects has been done using the activator IV for chiropractic care.
-Fuhr, A.W. The Activator Method. 2nd ed. St. Louis, MO: Mosby; 2008; 3-565. Fuhr, A.W. The Activator Method. 2nd ed. St. Louis, MO: Mosby; 2008; 3-565.
- What is the novelty of your finding? Previous finding by Niazi et, al (2015) mentioned that there is no change in H-reflex response but the H-reflex pathway can be infected by spinal manipulation.
Response: In the work of Niazi et al., (2015) traditional spinal manipulation was applied, with this technique more tissue is manually stimulated with the consequent activation of more afferent pathways. In our study, activator IV was used, in the first and tenth sessions we systematically recorded the H-reflex bilaterally in both soleus muscles, in order to explore all possible sources of differential effects. We can consider that it is a pioneering work that shows the lack of effects of the activator IV on the amplitude of the H-reflex, this does not mean that the clinical effects are not present when applying this technique. The subjective improvements reported by the experimental subjects were better motor control in the dance subjects and less fatigue in the non-dance subjects.
- Page 2, line 51, “Activator IV fixes the force at 0.3 J [1,2]”. Is it the maximum force you can reach or you use this force in your experiment to measure the H-reflex change. Also, the references are not from recent work and not involving the set-up of this work as well?
Response: The activator IV has 4 degrees of strength that must be selected according to the area to be treated:
|
Grade |
Force (lb) |
Adjustment area |
|
1 |
16 |
Cervical and pediatrics |
|
2 |
18 |
Thoracic and limbs |
|
3 |
21 |
Thoracic: middle and lowers |
|
4 |
38 |
Lumbar and pelvis |
In each experimental subject, the protocol indicated for each area to be treated was followed. In our experiments, the protocols in each session were executed by the same chiropractor, Brayan Martínez, who at that time had 2 years of international certification and extensive experience.
- Figure 1B-E, Why the dash lines are always higher from Average value, are the authors showing only the higher limit of standard deviation?
Response: for clarity of the graphs we are only showing the higher limit of standard deviation.
The corresponding text on the Figure legend now reads:
“For clarity of the H-reflex and M-wave graphs we are only showing the higher limit of standard deviation.”
- Figure 2 shows the peak amplitude and Hmax/Mmax ratio but the no. samples are not the same, it carries between 5-7, why?
Response: apparently, the number of points was different because many points were superimposed. Now we are showing the correct whisker plots, in which it is clear that the number of points is the same (7) for all the graphs.
- Overall, the Authors should performed some more relevant experiment using Activator IV for better understanding the H-reflex pathway by spinal manipulation.
Response: We believe that the experiments contained in this manuscript are necessary and appropriate to show the effects of the activator IV on the Hoffmann reflex. In fact, this is the first approach to documenting the electrophysiological effects of IV activator.
Minor:
- Double numbering of references
Response: now it's fixed.

Round 2
Reviewer 2 Report
Review responses are justified! This version of the manuscript can be accepted after some minor changes.
1. Include all the relevant references in the revised manuscript.
2. Include a modified Figure 1B-E with all your raw result in the supplementary sections.
Author Response
- Include all the relevant references in the revised manuscript.
Response 1: The relevant references have already been included.
- Include a modified Figure 1B-E with all your raw result in the supplementary sections.
Response 2. Raw data of both lower and upper values of standard deviation and all raw results are include in a new Figure 1 B-E and an excel file in the supplementary section, accordingly
